# Mesenchymal Stromal Cell-Derived Small Extracellular Vesicles Modulate Apoptosis, TNF Alpha and Interferon Gamma Response Gene mRNA Expression in T Lymphocytes

**DOI:** 10.3390/ijms241813689

**Published:** 2023-09-05

**Authors:** Andrea Fracchia, Drirh Khare, Samar Da’na, Reuven Or, Amnon Buxboim, Boaz Nachmias, Claudine Barkatz, Regina Golan-Gerstl, Swasti Tiwari, Polina Stepensky, Yuval Nevo, Hadar Benyamini, Sharona Elgavish, Osnat Almogi-Hazan, Batia Avni

**Affiliations:** 1Department of Bone Marrow Transplantation & Cancer Immunotherapy, Hadassah Medical Center, Jerusalem 9112001, Israel; andrea.fracchia01@gmail.com (A.F.); drirhkhare@gmail.com (D.K.); samardaana@gmail.com (S.D.); reuvenor@hadassah.org.il (R.O.); polina@hadassah.org.il (P.S.); osnath@hadassah.org.il (O.A.-H.); 2Faculty of Medicine, Hebrew University of Jerusalem, Jerusalem 9112001, Israel; nachmiasb@gmail.com; 3Department of Cell and Developmental Biology, Hebrew University of Jerusalem, Jerusalem 9190401, Israel; amnon.buxboim@mail.huji.ac.il; 4Department of Hematology, Hadassah Medical Center, Jerusalem 9112001, Israel; 5Department of Pediatrics, Hadassah-Hebrew University Medical Center, Jerusalem 9112001, Israel; reginag@hadassah.org.il; 6Department of Molecular Medicine & Biotechnology, Sanjay Gandhi Post Graduate Institute of Medical Sciences, Lucknow 226014, India; tiwari_pgi@yahoo.com; 7Info-CORE, Bioinformatics Unit of the I-CORE at the Hebrew University of Jerusalem, Jerusalem 9112001, Israel; yuval.nevo@mail.huji.ac.il (Y.N.); hadar.benyamini@gmail.com (H.B.); sharona.elgavish@mail.huji.ac.il (S.E.)

**Keywords:** mesenchymal stem cells, small extracellular vesicles, T lymphocytes, next-generation sequencing, Ingenuity Pathway Analysis, gene set enrichment analysis

## Abstract

Recent studies have highlighted the therapeutic potential of small extracellular bodies derived from mesenchymal stem cells (MSC-sEVs) for various diseases, notably through their ability to alter T-cell differentiation and function. The current study aimed to explore immunomodulatory pathway alterations within T cells through mRNA sequencing of activated T cells cocultured with bone marrow-derived MSC-sEVs. mRNA profiling of activated human T cells cocultured with MSC-sEVs or vehicle control was performed using the QIAGEN Illumina sequencing platform. Pathway networks and biological functions of the differentially expressed genes were analyzed using Ingenuity pathway analysis (IPA)^®^ software, KEGG pathway, GSEA and STRING database. A total of 364 differentially expressed genes were identified in sEV-treated T cells. Canonical pathway analysis highlighted the RhoA signaling pathway. Cellular development, movement, growth and proliferation, cell-to-cell interaction and inflammatory response-related gene expression were altered. KEGG enrichment pathway analysis underscored the apoptosis pathway. GSEA identified enrichment in downregulated genes associated with TNF alpha and interferon gamma response, and upregulated genes related to apoptosis and migration of lymphocytes and T-cell differentiation gene sets. Our findings provide valuable insights into the mechanisms by which MSC-sEVs implement immunomodulatory effects on activated T cells. These findings may contribute to the development of MSC-sEV-based therapies.

## 1. Introduction

Mesenchymal stem cells (MSCs) have been shown to secrete regulatory molecules and cytokines that dampen the proliferation, differentiation, maturation, migration, and function of immune cells [1,2,3,4]. The immunoregulatory effect of MSCs is mediated by both direct cell-to-cell contact and by paracrine effects, such as the release of multivesicular bodies (e.g., sEVs) [1,5,6,7,8,9,10,11]. Multivesicular bodies (EVs) are lipid bilayer-enveloped vesicles containing RNA molecules and proteins derived from the cell of origin. This heterogeneous vesicle population consists mainly of three types of vesicles: apoptotic bodies, ectosomes and sEVs, which differ in terms of biogenesis, size and enriched proteins. However, definitive distinctive features among these vesicle types have not been established [12,13]. sEVs, with a diameter ranging from 30 to 150, can be internalized by various cells through several pathways, including endocytosis, phagocytosis and micropinocytosis, enabling them to transfer their content to the recipient cell [14]. It has been observed that sEV-mediated transmission of RNA can affect mRNA expression and consequently protein production, suggesting an important role in intercellular communication [15,16]. Since bone marrow-derived MSCs (bmMSCs) can potentially serve as immunomodulators in various diseases [5,6,17,18], bmMSC-derived sEVs (bmMSC-sEVs) hold promise as therapeutic candidates for immune-associated diseases such as inflammatory bowel disease (IBD), autoimmune disease and stem cell-associated graft-versus-host disease (GVHD) [19,20,21,22]. We have previously demonstrated that sEVs modulate mRNA expression in B cells, leading to reduced B cell proliferation and decreased IgM production [23]. Several studies have demonstrated that MSC-derived sEVs induce apoptosis in activated T cells, and specific subpopulations [8]. They have also been shown to increase the percentage of T naïve cells and decrease the percentage of CD4/CD8 effector T cells [24], and to reduce levels of IL-2, TNF alpha and INF-gamma in mouse models of acute GVHD [25]. Therefore, in the current study, we aimed to investigate the differential expression levels of activated mRNA in T cells cocultured with bmMSC-sEVs and identify the key pathways influenced by these expression changes drawing connections to existing in vitro and in vivo findings, suggesting potential clinical implications.

## 2. Results

### 2.1. bmMSC-sEV Characterization

bmMSC-sEVs were isolated by using serial differential centrifugations. Three batches of sEVs were isolated from three different bmMSC donors. Electron microscopy imaging established the presence of isolated bmMSC-sEVs (Appendix A). Nanoparticle tracking analysis (NTA) was used in order to evaluate the average size of the bmMSC-derived sEVs. The size range of the MSC-derived microparticles was 90–120 nm. Western blotting established positive expression of CD63 and annexin V and negative expression of KDEL on these bmMSC-derived sEVs, confirming MSC-sEV protein identity.

### 2.2. bmMSC-sEVs Inhibit T Cell Proliferation

In our previous study [23], we found a significant inhibition of proliferation in the presence of bmMSC-sEVs isolated from 1 × 10^6^ MSCs from various donors. Thus, in our current experiments, we used three batches of sEVs isolated from 1 × 10^6^ MSCs obtained from three separate donors. T cells isolated by an Easysep^TM^ human T-cell enrichment cocktail and activated with CD3/CD28 were incubated with and without bmMSC-sEVs. We observed a 26.2% reduction in proliferation, as assessed by thymidine incorporation assay (*p* ≤ 0.01) (Figure 1a).

### 2.3. sEV Internalization by Activated T Cells

Internalization of bmMSC-sEVs by activated T lymphocytes was assessed by confocal microscopy. sEVs were labeled with acridine orange dye, Figure 1b–e shows a colocalization of the dye in T cells cocultured with bmMSC-sEVs and a negative signal in T cells cocultured with the dye only, supporting uptake of sEVs by activated T lymphocytes.

### 2.4. mRNA Expression Profile of T Cells Incubated with bmMSC-sEVs

mRNA was extracted from activated T lymphocytes incubated with and without bmMSC-sEVs (isolated from 3 separate bone marrow donors). Using the Illumina sequencing platform, mRNA profiling was performed, and differential expression (DE) genes were calculated using DESeq2. mRNA profiling of activated T lymphocytes revealed 364 DE genes (padj < 0.1) between sEV-treated and control cells (Appendix A and Figure 2). In the upregulated group, we found proapoptotic genes, such as TNFRSF8 and TP53, genes that are mainly overexpressed in naïve and memory T cells, such as CD7, CD27, CCR7, and LTA, a member of the tumor necrosis factor family. In contrast, in the downregulated group, we observed downregulation of cytokines, such as IL2 and IL23A, and the antiapoptotic gene MCL1.

### 2.5. Cutoff Analysis

Qiagen Ingenuity Pathway Analysis (IPA)^®^ was applied to the genes that were significantly DE (padj < 0.1). Canonical pathway analysis showed enrichment (−log (B-H *p*-value) > 1.3) of the following pathways: RhoA signaling and regulation of actin-based motility by Rho, caveolar-mediated endocytosis signaling, cross talk between dendritic cells and natural killer cells, integrin signaling, Huntington’s disease signaling, clathrin-mediated endocytosis signaling, and RhoA signaling and phagosome maturation (Figure 3a, Appendix A). An IPA^®^ schematic diagram of canonical signaling pathway regulation of actin-based motility by Rho showing the relative gene expression changes in T cells cocultured with MSC-derived sEVs is presented in Figure 3b.

Qiagen IPA^®^ function and disease analysis showed enrichment of several categories, including cellular movement, development, growth and proliferation, cell death and survival, cell mediated immune response, inflammatory response, immune cell trafficking, cell-to-cell signaling and interaction, and cell signaling (Appendix A). Looking at disease and function annotation analysis revealed a Qiagen IPA^®^ z score < −2 for the differentiation of hematopoietic progenitor cells and cell proliferation of tumor cell lines (Appendix A), implying a significant decrease in both functions. Using DE genes relevant to immune cell-to-cell signaling and interaction (Appendix A), we created a molecular network, choosing only direct and experimental connections (Figure 4).

Enrichment analysis of KEGG pathways using the STRING database revealed the apoptosis pathway to be the most relevant affected pathway. The pathway is displayed with overlay of genes from the DE gene list in Figure 5.

### 2.6. Non-Cutoff Analysis

Running the GSEA against the hallmark gene set collection from the molecular signature database (MSigDB), we found an enrichment in downregulated genes related to TNF alpha and interferon (INF) gamma response in the sEV-treated samples (Figure 6a,b). These results were further supported by running our DE genes on the meta-server Enricher [26,27,28] showing that MSigDB Hallmark was enriched (adjusted *p* value < 0.05) for INF gamma response, INF alpha response, and IL2/STAT5 signaling gene sets. Applying our samples to the collection of T-cell-related gene sets from Qiagen IPA^®^, we found a significant enrichment in upregulated genes related to apoptosis of lymphocytes, lymphocyte migration, and T-cell-differentiation gene sets (Figure 6c–e).

## 3. Discussion

Human MSCs were shown by several studies to regulate human mononuclear cell proliferation in a dose-dependent manner [2,3,4]. Although the exact mechanism is still not fully understood, it is likely that the complex activity of MSCs affects various immune cell types, including T cells. These cells have been shown to impact T-cell subpopulation ratios, induce apoptosis in activated T cells, upregulate the indoleamine 2,3-dioxygenase (IDO)-mediated tryptophan catabolism pathway and transmit their effects through multiple soluble factors [29,30,31]. Several studies demonstrated that EVs, and in particular sEVs, play a role in these transmission mechanisms by modulating mRNA expression in immune cells [5,6,7,8,9,32]. Furthermore, they have been shown to regulate several key biological properties in both B and T cells [8,24,25,33]. In our previous work, we demonstrated that bmMSC-sEVs can suppress proliferation of PBMCs, particularly B and T cells. B cells cocultured with bmMSC-sEVs resulted in differential expression of 186 genes involved in cell trafficking, development, hemostasis, and immune cell function. Our present study aimed to investigate the changes in gene expression and the affected pathways when coculturing T cells with bmMSC-sEVs. mRNA profiling of activated T lymphocytes revealed 364 DE genes between bmMSC-sEV-treated and control cells.

MSC-derived EVs have exhibited the ability to induce apoptosis in various cell types while inducing the opposite effect in others, depending on several factors, such as the source of the MSCs [8,34,35,36]. Both Mokarizadeh et al. and Del Fattore et al. have demonstrated that unlike MSCs, MSC-EVs increased apoptosis in the overall CD3^+^ population [8,37]. Furthermore, sEVs derived from IL-1β-stimulated as well as unstimulated MSCs were shown to induce apoptosis in activated T cells, as well as in CD3, CD4 and Treg cells [37]. In our research, we observed upregulation of proapoptotic genes TNFRSF8 and TP53 along with downregulation of MCL1, which is known to promote survival in activated T cells [38]. Utilizing Qiagen IPA^®^ function and disease analysis, we identified enrichment of genes involved in intercellular signaling and interaction (Appendix A). Based on the DE genes participating in these functions, a network map was constructed (Figure 4). This network highlights the central role of upregulated TP53 in T-cell regulation by bmMSC-sEVs. Several studies have demonstrated that TP53 upregulation in T cells inhibits proliferation by inducing senescence, causing cell cycle arrest in the G2/M phase and promoting apoptosis [39,40,41].

We have found upregulation of genes that are mainly overexpressed in naïve and memory cells, such as CD7, CD27 and CCR7. CD7 is a transmembrane glycoprotein that appears early in T-cell ontogeny and highly expressed in naïve and memory cell subsets [42]. CD27, a member of the TNFR family, has been used to define various stages of T-cell differentiation and is expressed by naïve CD4 and CD8 T cells and most memory T cells [43,44]. CCR7 is expressed at high levels on naïve and central memory T cells with minimal capacity to secrete cytokines and enables T-cell homing to lymphoid organs [43]. Furthermore, Qiagen IPA^®^ shows a significant decrease in differentiation of hematopoietic and progenitor cell function. Fujii et al. demonstrated that MSC-derived EV-treated GVHD mice show a decreased rate of CD4/CD8 effector T cells with an increase in naïve T cells [24]. Our results support this functional evidence showing bmMSC-sEVs suppress differentiation of T cells from a naïve phenotype to an effector phenotype.

Canonical pathway analysis has shown enrichment of the Qiagen IPA^®^ RhoA signaling pathway and regulation of actin-based motility by the Rho pathway in the sEV-treated cells (Figure 3a). Rho-GTPases are crucial signal transducing proteins. Activated GTP-bound Rho-GTPases interact with various other effectors, regulating multiple cellular pathways that affect cytoskeletal dynamics, motility, migration, cell growth and proliferation, apoptosis, gene expression and nuclear signaling. In T lymphocytes, Rho-GTPases have been shown to regulate fundamental functions including cell division, migration, activation and adhesion. Notably, in our model, based on the IPA^®^ canonical pathway analysis, ROCK (a downstream effector of the small GTPase RhoA) was predicted to be downregulated (Figure 3b). ROCK2 has been shown to be upregulated in Th17 cells, while its inhibition tips the balance toward Treg cells. Recent studies have highlighted the potential of ROCK2 inhibition as a promising therapeutic approach for immune-associated diseases such as systemic lupus erythematosus (SLE), IBD, and chronic GVHD [45,46,47]. In addition, actin regulatory proteins including ADF/cofilin have been shown to play a crucial role in apoptosis regulation [48,49,50]. Our findings indicate an upregulation of cofilin 1 (Figure 3b), a key modulator of actin, in activated T cells treated with MSC-derived sEVs, thus emphasizing the role of sEVs in cell motility, migration and adhesion. Furthermore, canonical pathway analysis showed caveolar-mediated endocytosis signaling to be affected (Figure 3a). Caveolar-mediated endocytosis signaling has been recognized as an important pathway for the cellular uptake of sEVs [51], implying that sEVs may upregulate their own internalization into T cells.

Correlating with the in vitro decreased proliferation of T cells, there was downregulation of proinflammatory cytokines such as IL2 and IL23A, as well as genes related to TNF alpha and interferon gamma response (Appendix A, Figure 4 and Figure 6a,b). These findings align with the conclusions drawn by Ke-Liang Li et al., showing decreased levels of IL-2, TNF alpha and INF-gamma in both human in vitro and mice in vivo acute-GVHD models after treatment with MSC-derived sEVs [25]. These findings support the use of sEVs for the treatment of immune associated diseases such as GVHD, where IL2, TNF alpha and interferon gamma play a major role.

The major limitation of our study stems from its in vitro design. In order to assess the actual impact of MSC EVs on T-cell immunity, further studies involving several additional T-lymphocyte donors as well as human in vivo studies are needed. Furthermore, we used only one set of T cells from a single donor for the mRNA expression profiling. This approach was adopted to minimize alterations in the experimental conditions and maintain consistency. Future studies should address the variability ingrained in T cells from different donors. Human studies are still lacking, but mouse studies have provided evidence consistent with our in vitro findings predominantly indicating decreased levels of IL-2, TNF alpha and INF gamma [24,25,52]. As there is growing interest in exploring the potential therapeutic applications of MSC-derived EVs for immune-associated diseases, it is crucial to gain a deeper understanding of the underlying mechanisms by which these vesicles exert their immunomodulatory effect. The present study adds valuable insight to our growing understanding of the impact of MSC-derived EVs on T-cell behavior by exploring the impact of these vesicles on T-cell mRNA expression.

## 4. Materials and Methods

### 4.1. MSC Separation and Expansion

Bone marrow (BM) aspirates were obtained from the iliac crest of three healthy human donors for allogeneic transplantation, upon approval of the local Helsinki Committee (HMO-0626-15). All participants gave written informed consent in accordance with the Declaration of Helsinki (participant characteristics are presented in Appendix A). BmMSCs were separated from the BM aspirates by filtration through nylon cell strainers with 70 µm pores (SPL Life Sciences, Pocheon-si, Gyeonggi, Republic of Korea) and resuspended in filter-sterilized Dulbecco’s modified Eagle medium (DMEM, Biological Industries, Beit-Haemek, Israel) supplemented with 15% heat-inactivated fetal bovine serum (HI-FBS, Biological Industries, Beit-Haemek, Israel), 1% L-glutamine (200 mM) (L-Glu, Biological Industries, Beit-Haemek, Israel), and 1% penicillin (100 units/mL)–streptomycin (100 μg/mL)–neomycin (100 μg/mL) (P/S/N, Biological Industries, Beit-Haemek, Israel). Cells were plated and cultured at 37 °C in a humidified atmosphere with 5% CO_2_. For cell passaging, plates approaching ~80% cell confluency were incubated for 2–3 min in 5 mL of prewarmed trypsin EDTA solution (0.25% trypsin, 0.05% EDTA, Biological Industries, Beit-Haemek, Israel) at 37 °C and then plated in fresh growth medium. During passages 1–3, when cells reached 80–90% confluency, they were suspended for 24 h in growth medium supplemented with sEV-depleted HI-FBS. Cell identity was confirmed by flow cytometry analysis (FACS) using established criteria with positive markers (anti-CD73, CD166, CD105, HLA-ABC and CD90 antibodies) and negative markers (anti-HLA-DR, CD56, CD3 and CD45 antibodies) (Biolegend, San Diego, CA, USA) (Appendix A).

### 4.2. PBMC Isolation

Peripheral blood was drawn from several healthy donors under approval of the local Helsinki Committee (HMO-0513-11). Donor characteristics are presented in Appendix A. Mononuclear cells were obtained by centrifugation over a Lymphoprep^TM^ density gradient (Stem Cell^TM^ Technologies, Vancouver, BC, Canada). After collecting the buffy coat, cells were washed twice in phosphate-buffered saline (PBS, Biological Industries, Beit-Haemek, Israel) and plated in RPMI 1640 (Biological Industries, Beit-Haemek, Israel) supplemented with 10% sEV-free HI-FBS at a concentration of 1 × 10^6^ cells/mL.

Each individual experiment was performed using T cells isolated from a single donor. The reason behind this approach was to maintain consistency within each experiment and avoid introducing variability by using T cells from multiple donors.

### 4.3. BmMSC-Derived Small EV Isolation, Purification, and Characterization

Small EV isolation and purification has been previously described [23]. Briefly, medium was collected and differentially centrifuged at 300× *g* for 10 min at room temperature (R.T.) and at 10,000× *g* for 20 min at RT to eliminate cells and debris. The supernatant was filtered through 0.22 µm filter bottles (Bar-Naor Ltd., Petah Tikvah, Israel) and ultracentrifuged at 100,000× *g* for 1 h at 4 °C using a SW 28 rotor in an Optima L-90K ultracentrifuge (Beckman Coulter, Brea, CA, USA). sEV pellets were resuspended in PBS and ultracentrifuged again at 100,000× *g* for 1 h at 4 °C. sEVs were finally resuspended in PBS and stored at −80 °C for further analysis. Each batch of sEVs was isolated from a different MSC donor.

Electron microscopy was performed using transmission electron microscopy (Appendix A), Particle size was determined using the nanoSight nanoparticle tracking analysis (NTA) system (Malvern Instruments, Malvern, UK), Western blotting using positive (annexin-V, CD63, flotillin 1) and negative (KDEL) markers, and FACS analysis for CD63 and CD81 as previously described [23].

### 4.4. T-Cell Isolation and Activation

In vitro T-cell isolation using the Easysep^TM^ human T-cell enrichment cocktail was performed according to the manufacturer’s protocol (Stem Cell^TM^ Technologies, Vancouver, BC, Canada). Briefly, 50 µL of Easysep™ human T-cell enrichment cocktail was added to 5 × 10^7^ cells re-suspended in 1 mL PBS containing 2% FBS and 1 mM EDTA. After a short incubation, 40 µL/mL of Easysep™ RapidSpheres™ were added and the tube was placed in a magnet. Isolated T cells were collected and counted.

T-cell activation was performed by covering designated wells with 50 µL of 5 µg/mL anti CD3 (UCHT1) antibody (Tonbo Biosciences, San Diego, CA, USA), followed by incubation at 37 °C for 4 h. Antibody solution in each well was replaced by 100 µL 10% sEV-free RPMI media for blocking. T cells were added at a concentration of 1 × 10^5^ cells per well and cultured with 0.4 µg per well of anti CD28 (CD 28.2) antibody (Tonbo Biosciences, San Diego, CA, USA) in the presence or absence of purified bmMSC-sEVs (isolated from 1 × 10^6^ MSCs, from three different donors), supplemented with 0.2 mL sEV-depleted medium (containing RPMI with 10% sEV-depleted FBS). After 4 days of culture, cells were collected for RNA extraction. For proliferation assays, cells were pulsed for 16 additional hours with 3H-thymidine at 1 µCi/well (PerkinElmer, Boston, MA, USA) and harvested. Top Count NXT (PerkinElmer, Buckinghamshire, UK) was used to assess 3H-thymidine incorporation. Proliferation and RNA extraction experiments were performed with a single PBMC donor.

### 4.5. Labeling of sEVs

sEV RNA was labeled using acridine orange (Invitrogen, Carlsbad, CA, USA). We added 1 μL of acridine orange (10 mg/mL) to 100 μL of PBS containing sEVs, derived from 5 × 10^5^ MSCs. After thorough mixing, the sample was incubated for 10 min at 37 °C. To halt the labeling process, 30 μL of ExoQuick-TC reagent (System Biosciences, Palo Alto, CA, USA) was added to the labeled sEV suspension, and the mixture was inverted six times. The sample was then placed on ice for half an hour, after which it was centrifuged at 14,000× *g* for 3 min to sediment the vesicles.

### 4.6. Internalization Assay

The pellet, containing the labeled sEVs, was resuspended in PBS and incubated with 5 × 10^5^ T lymphocytes for two hours and then fixated to a coverslip. For the negative control, we employed free acridine orange at a concentration of 1 μg in 100 μL of PBS. Images were collected using a Nikon A1R+ confocal microscope (Nikon Corporation, Tokyo, Japan) and Zeiss LSM 980 laser (Carl Zeiss Meditec Group, Jena, Germany) scanning confocal microscope.

### 4.7. Lymphocyte RNA Extraction and Library Preparation

Lymphocyte RNA was extracted using the trizol method. For quality control of RNA extraction yield and library synthesis products, an RNA Screen Tape kit (Agilent Technologies, Waldbronn, Germany), D1000 ScreenTape kit (Agilent Technologies, Waldbronn, Germany), Qubit^®^ RNA HS Assay kit (Invitrogen, USA) and Qubit^®^ DNA HS Assay kit (Invitrogen, USA) were used for each specific step. For mRNA library preparation, poly-A selection beads followed with NEXTflex™ Rapid Directional qRNA-Seq™ Kit were used (Bio Scientific, Cambridge, UK). In brief, 1 μg was used for the library construction and the library was eluted in 20 μL elution buffer. Libraries were adjusted and pooled to a concentration of 4 nM. Multiplex sample pools including PhiX 1.5% were loaded on a NextSeq 500/550 High Output v2 kit (75 cycles) cartridge (Illumina, San Diego, CA, USA) and loaded on the NextSeq 500 System (Illumina, USA), with 75 cycles and single-read sequencing condition.

### 4.8. Mapping and Differential Expression Analysis

RNA-seq data were analyzed as previously described [23]. In short, raw reads were processed with cutadapt, v1.18, to remove low-quality and adapter sequences. Processed reads were aligned to the human genome with tophat, v2.1.1, and raw counts were calculated with htseq-count, v0.6.0. The raw counts were normalized, and differential expression was calculated with the R package DESeq2, v1.22.2.

### 4.9. Target Prediction and Pathway Analysis

**Cutoff analysis:** Using the cutoff analysis approach, we searched for enriched pathways in the list of differentially expressed (DE) genes. This approach allows a focus on the genes that display significant changes in expression and facilitates the investigation for known biological pathways that are statistically enriched within this subset. Genes were defined as DE in activated lymphocytes incubated with bmMSC-sEVs in comparison with nonincubated lymphocytes based on false-discovery rate (FDR) corrected *p*-value < 0.1, obtained by using DESeq2 Wald test.

The list of DE genes was subjected to enrichment analysis (canonical pathways, molecular functions and diseases) using Ingenuity Pathway Analysis (IPA^®^, Qiagen Inc., redwood City, CA, USA; https://digitalinsights.qiagen.com/products-overview/discovery-insights-portfolio/content-exploration-and-databases/qiagen-ipa/) [53]. In addition, the DE list was subjected to hypergeometric analysis in the STRING database [54]. Selected enriched KEGG pathways (Kyoto Encyclopedia of Genes and Genomes (KEGG) [55] pathways (FDR < 0.05) were depicted with overlay of genes from the DE list.

**Non-cut-off analysis:** As a complementary approach, we also conducted an analysis of the entire expression data. This approach is more attuned to detecting mild expression alterations and is not biased by an arbitrary cutoff parameter. The complete gene expression data set was subjected to gene set enrichment analysis using the Broad Institute’s GSEA tool [56]. GSEA utilizes all differential expression data to determine whether a priori-defined set of genes show statistically significant concordant differences between two biological states. GSEA was run against the hallmark gene set collection from the molecular signature database (mSigDB), as well as a collection of T-cell-related gene sets extracted from Qiagen IPA^®^. The genes were ranked according to the extent of alterations in their expression when comparing activated lymphocytes that were incubated with bmMSC-sEVs and those that were not.

## 5. Conclusions

Our findings contribute to the understanding of the mechanisms by which bmMSC-sEVs implement the immunomodulatory properties of their parent cells. Specifically, in the case of T-cell immunity, we observed a significant effect on the RhoA signaling pathway, which plays an important role in various key processes of T-cell development, activation, differentiation and migration, upregulation of proapoptotic genes, upregulation of genes that are mainly overexpressed in naïve and memory cells, and downregulation of the IL2 gene and genes related to TNF alpha and interferon gamma response.

## Figures and Tables

**Figure 1 ijms-24-13689-f001:**
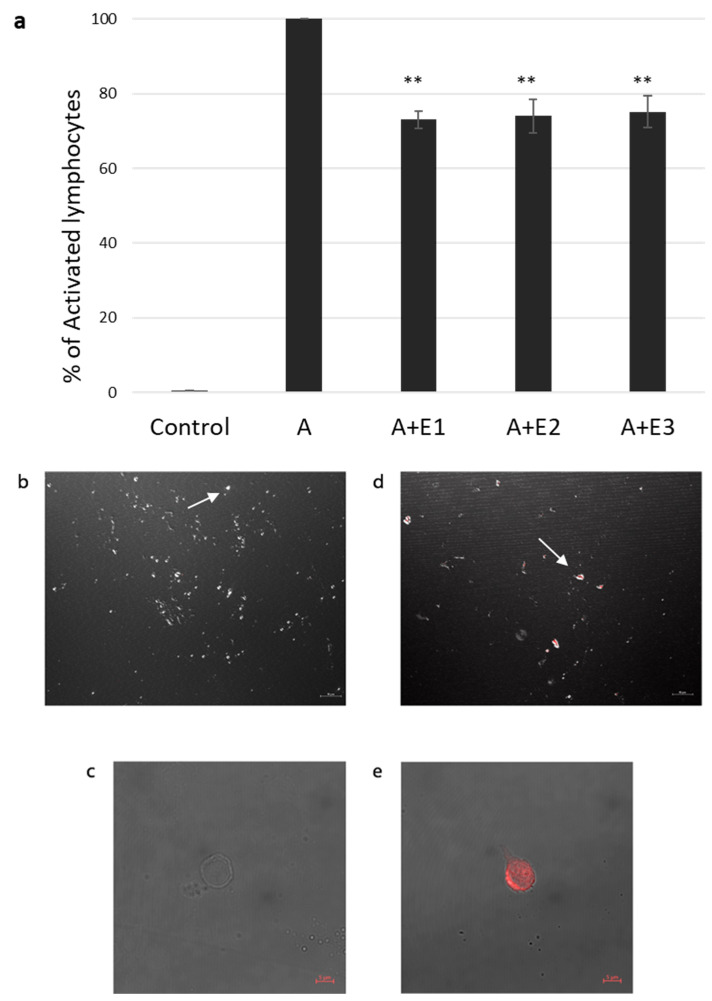
sEV suppression of activated T lymphocytes and internalization assay. (**a**). Thymidine incorporation assay. Three different batches (E1, E2, E3) of mesenchymal stem cell (MSC)-derived sEVs cocultured with CD3/CD28-activated (A) single donor T lymphocytes. Data is represented as mean ± SE. The mean was calculated from 4 independent experiments. Control: nonactivated lymphocytes. Activated condition counts were set to represent 100%, ** *p* < 0.01. (**b**–**e**). sEV internalization by activated T lymphocytes was assessed using confocal microscopy. sEV RNA was labeled using acridine orange dye. (**b**,**c**). Negative control: T cells incubated for two hours with the dye (without the sEVs). (**b**). Magnification 40×, scale bar = 10 µm. Arrow pointing at an unstained cell (**c**). Magnification 64×, zoom 2×, scale bar = 5 µm. (**d**,**e**), T cells incubated for two hours with labeled sEVs (red). (**d**). Magnification 40×. Scale bar = 10 µm. Arrow pointing at a T cell containing the stained sEVs (**e**). Magnification 64×, zoom 2×, scale bar = 5 µm.

**Figure 2 ijms-24-13689-f002:**
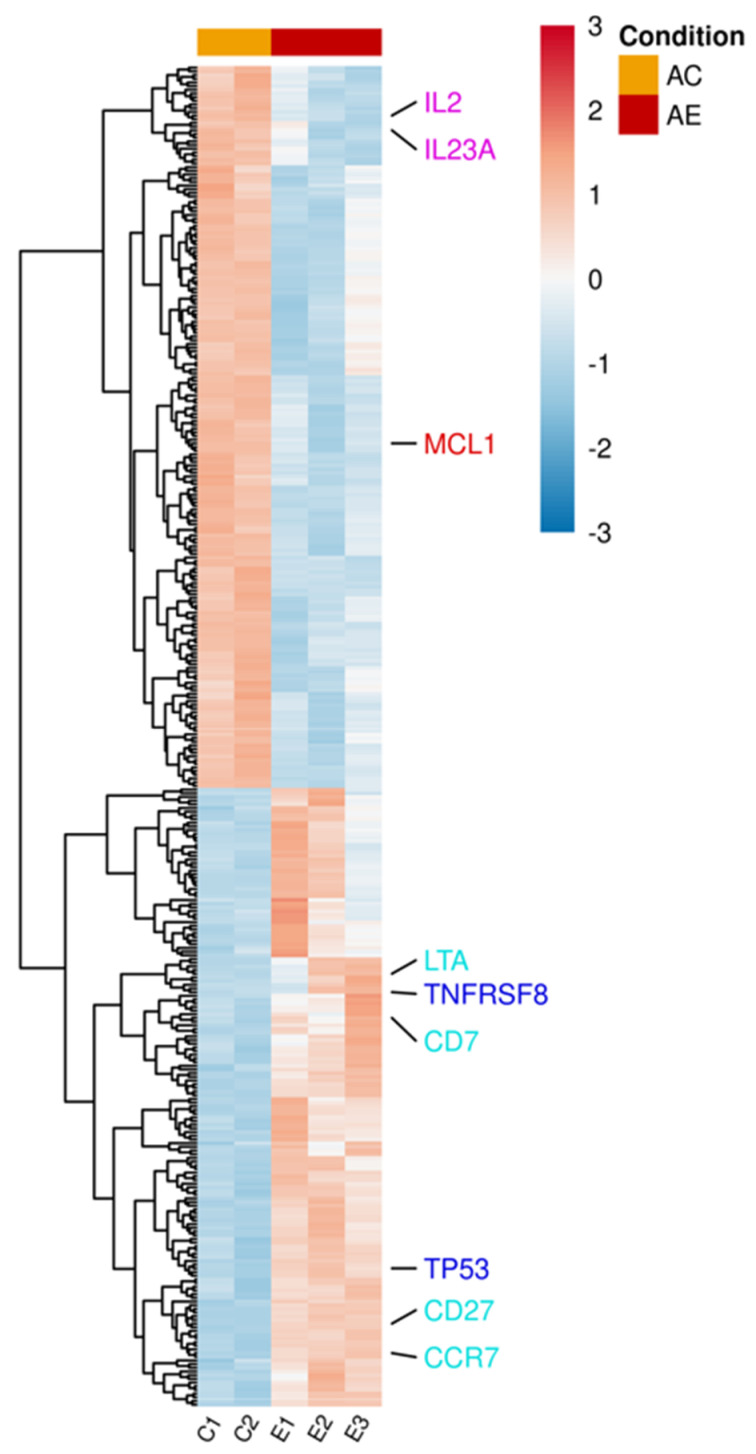
Differentially expressed genes in activated T lymphocytes incubated with sEVs. Single donor activated lymphocytes cocultured with three different batches of bmMSC-sEVs (E1, E2, E3) obtained from three different bone marrow donors were compared to a duplicate of activated lymphocytes (C1, C2). The normalized expression of 364 significantly expressed genes, padj < 0.1, is shown as a heatmap, after scaling the values for each gene. The color scale is indicated on the top right corner (blue, below average; red, above average). Genes are ordered by hierarchical clustering. Genes associated with pathways discussed in the manuscript were color-coded for clarity. Proapoptotic genes are highlighted in blue, T-cell differentiation genes in sky blue, cytokine genes in purple, and antiapoptotic genes in red. AC, activated T cells; AE, activated T cells cocultured with sEVs.

**Figure 3 ijms-24-13689-f003:**
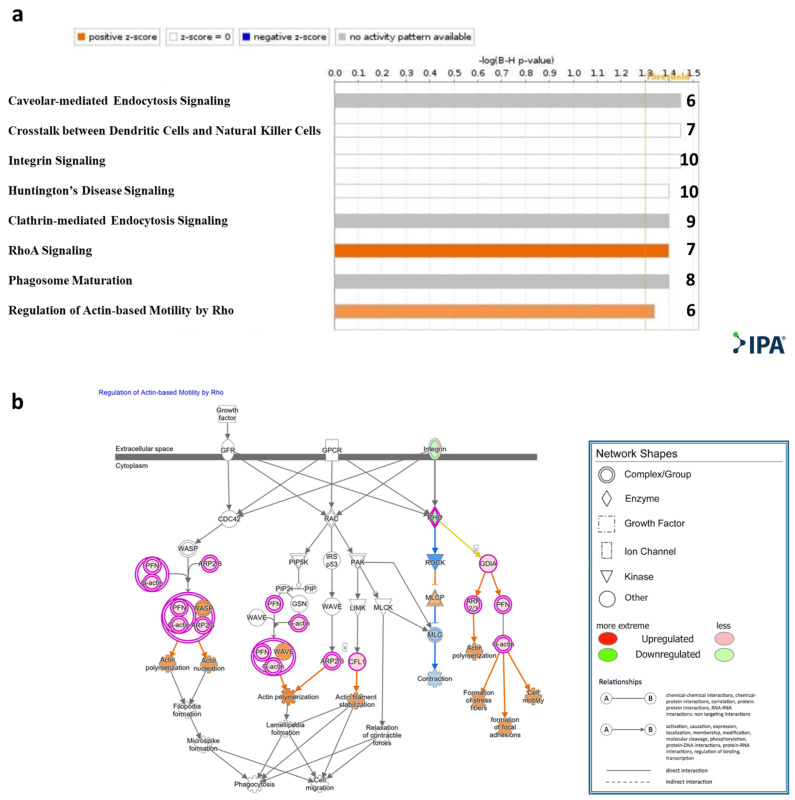
IPA^®^ Canonical pathway analysis. (**a**). Statistically significant enriched canonical pathways (Y axis) listed according to their B-H *p*-values (X axis; −log(B-H *p*-value) = 1.3, orange line). Orange bars: positive z-score; gray bars: no activity pattern available. On the right of each bar: number of overlapping DE genes in T cells cocultured with sEVs participating in each depicted canonical pathway. (**b**). Qiagen IPA^®^ regulation of actin-based motility by Rho pathway. Experimental upregulated (pink) and downregulated (green) DE genes in T cells cocultured with MSC-derived sEVs. Predicted upregulated and downregulated by the molecular activity predictor are depicted in orange and blue, respectively.

**Figure 4 ijms-24-13689-f004:**
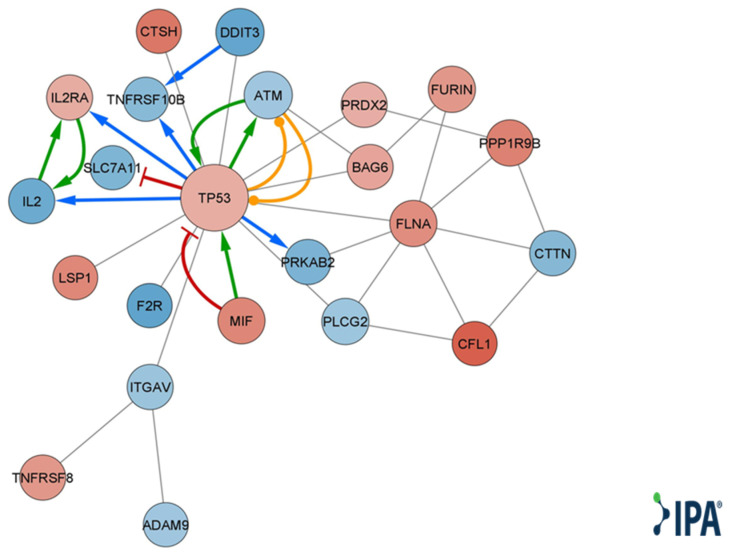
Qiagen IPA^®^ function and disease analysis gene interaction network map. A molecular network of direct and experimental interactions of DE genes relevant for cell-to-cell interaction. Nodes represent DE genes participating in the network. Node size corresponds to the number of neighboring nodes. Blue nodes represent lower expression levels in T cells cocultured with MSC-derived sEVs. Red nodes represent higher expression levels in T cells cocultured with MSC-derived sEVs. Edges represent cross talk between the DE genes. Red edge corresponds to inhibition, green edge to activation, orange edge to phosphorylation, blue edge to transcription, gray edge to protein–protein interaction, protein–DNA interaction, or expression.

**Figure 5 ijms-24-13689-f005:**
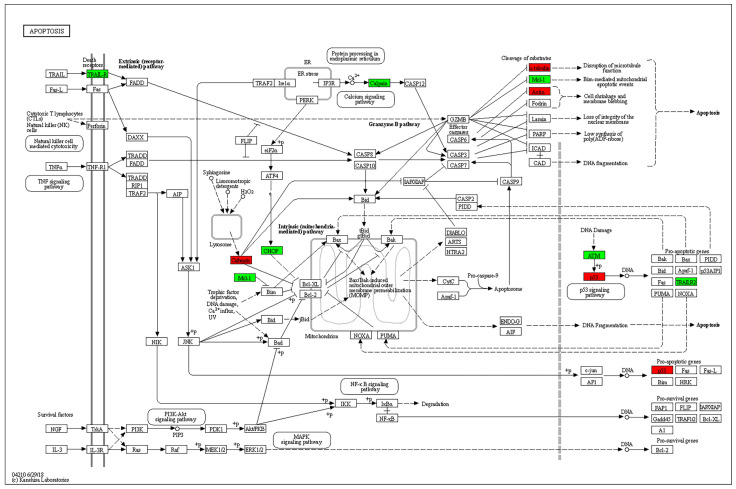
KEGG pathway analysis. The apoptosis pathway is displayed with overlay of genes from the DE gene list. Green: experimental downregulated genes. Red: experimental upregulated genes in T cells cocultured with MSC-derived sEVs.

**Figure 6 ijms-24-13689-f006:**
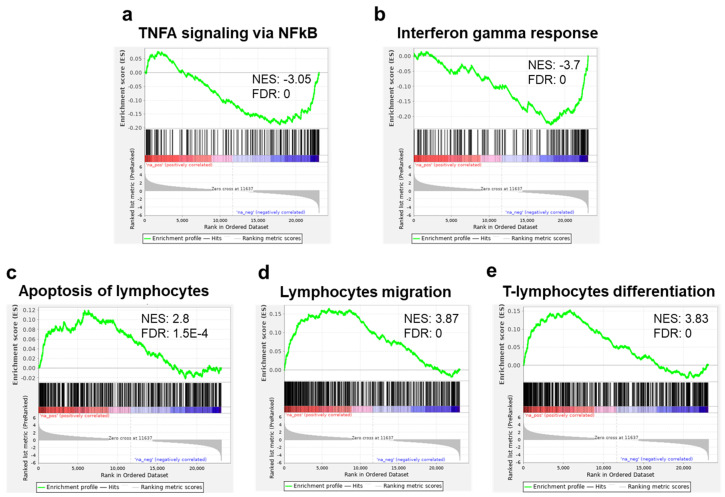
Gene set enrichment analysis (GSEA): Regulation of inflammatory related and immune gene sets. Enrichment plots for TNFA signaling via NFKB (**a**) and interferon gamma response (**b**), showing a significant reduction of expression (enrichment in the downregulated genes) of these gene sets in T cells cocultured with MSC-derived sEVs. Enrichment plots of genes in apoptosis of lymphocytes, (**c**) lymphocyte migration, and (**d**) T-cell lymphocyte differentiation (**e**), showing significant increase in expression (enrichment in the upregulated genes) of T cells cocultured with MSC-derived sEVs. Immune gene sets (**c**–**e**) were extracted from IPA^®^.

## Data Availability

Data is contained within the article or Appendix A.

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
