# Peer review of "Mesenchymal Stromal Cell-Derived Small Extracellular Vesicles Modulate Apoptosis, TNF Alpha and Interferon Gamma Response Gene mRNA Expression in T Lymphocytes"

_ijms, 2023, doi:10.3390/ijms241813689_

Round 1

Reviewer 1 Report

The purpose of the study was to clarify the mechanisms by which bmMSC-sEVs implement the immunomodulatory properties of their parent cell. Authors found DE genes in sEVs treated T-cells using the QIAGEN 25 Illumina sequencing platform and evaluated pathway networks and biological functions of the DE genes using IPA® software, KEGG pathway, GSEA and STRING database. The analysis showed downregulated genes associated with TNF alpha and Interferon gamma response, while upregulated genes related to apoptosis and migration of lymphocytes and T-cell differentiation gene sets. These results support the previous study on functional evidence showing bmMSC-sEVs suppresses differentiation of T cells from a naive phenotype to an effector phenotype. This network highlights the central role of upregulated TP53 in T cell regulation by bmMSC-sEVs.

It’s meaningful to demonstrate the mechanisms of immunomodulatory effects of sEVs on immune cells.

P3, line 106 Mark sEVs with arrows in Merge panel. 

Reviewer 2 Report

Dear authors,

the work should be implemented before the publication, the results obtained are interesting but the way the authors presented the results are not so satisfying.

Below some suggestion to improve the paper.

First of all the authors should put a big attention to the acronym used: EV is extracellular vesicles and no multivesicular that are annotated as MVB. because multivesicular bosies are specialized endosomes that contain intraluminal vesicles generated from invagination and budding of the limiting membrane as reported by Xifeng Li et al. 2018 and some others.

In the paper 12-14 they defined some criteria to establish the the different types of vesicles probably is better to put some other references.

For the characterization of the EV is mandatory to characterize them also for other markers typical of EVs, and also use another technique for the characterization such as FACS. From the MISEV guidelines the authors need at least two techniques to demonstrate the EVs characterization.

In the figure 1 is better to  report the statistical analysis.

In the paragraph 2.3: 1. How the authors demonstrate that there was an internalization of the sEVs, from the images proposed it seems that the vesicles were outside of the cells. I suggest also to perform a fluorescence staining for actin to visualize the internalization of the sEVs. 2. How can the authors should demonstrate that the PKH67 is specific for the uptake of sEVs. They performed a negative control to exclude the aspecific uptake?

In the figure 2 the authors should indicate which was the AC and AE conditions. Still in figure 2 the other genes that are not included in DE genes, were implicated in different role or they are not specific for a function? Why the authors select only gene involved in cell death, inflammatory response and development?

In figure 6 probably, linked to this graph, is better to put the second part of the graph indicating the Ranked list metric, to show which is positively correlated and negatively correlated

In the discussion at page 8 what the authors means with activated and non-activated MSC?

In the M&M section specify which was the concentration of Glu, P/S/N used, the percentage is not enough

In the supplementary material is better to show some FACS result regarding the characterization of bm-MSC.

The authors used the PBMC derived from how many donors? Did they mixed the PBMC derived from the donors? Why after they write that they used PBMC derived from one donor?

Briefly describe the T cells isolation and elucidate how many PBMC were needed.

How much time the authors stimulate the cells with the sEVs? For all the four days? How the authors chose 4 days of incubation if after 24h see the internalization of the sEVs?

For sEVs labeling why the authors did not put also a control with only PBS and the PKH67 green to demonstrate the specific labeling and uptake?

EVs internalization assay and immunostaining: Rewrite this paragraph because is not so clear how was performed the immunofluorescence. The cells were cultivated 24h with the sEVs and after why they was detached form the plate to plate again in a glass coverslips

Only some minor editing of English language are required

Reviewer 3 Report

Generally a well written paper. I have only one qualification and that is that only a single donor of the peripheral blood monocytes was used. The results, therefore, apply only to a single individual, even if the MSc derived EVs came from three donors. This limitation needs to be emphasised and discussed; and the Conclusions modified appropriately.

Minor errors:

 Should it be “MSC-derived sEVs” rather than “MSCs-derived sEVs” everywhere in the Discussion, the double plural is an unusual construct. “bmMCS-sEVs” is used as the relevant term in the Results

Also “sEV ” in the Abstract and Introduction Lines 28, 36 and 57, 92, 100

Line 54: “… from 30 to 150 nm…”

Line 65: “ MSC derived

Line 89: not enough data points to quote proliferation reduction to 3 significant figures.( use 30%)

Line 114: “sEV-treated”

Line 242: significant is missing the ‘t’

Line 282: “… potential therapeutic applications …” perhaps

Line 399: use actual reference

English expression is very good. Only minor comments on use of "EVs" and "MSCs" plurals as above.
